Identification and characterization of expression profiles of neuropeptides and their GPCRs in the swimming crab, Portunus trituberculatus

Tu Shisheng
Xu Rui
Wang Mengen
Xie Xi
Bao Chenchang baochenchang@nbu.edu.cn baocc203@163.com
Zhu Dongfa zhudongfa@nbu.edu.cn
School of Marine Science, Ningbo University , Ningbo , Zhejiang , China
Waiho Khor
Electronic publication date: 2021 Sep 15
Publication date: 2021
Volume: 9
Electronic Location ID: e12179
Received 2021 May 31; Accepted 2021 Aug 29
Copyright: ©2021 Tu et al.
Copyright year: 2021
Copyright holder: Tu et al.
License: This is an open access article distributed under the terms of the Creative Commons Attribution License, which permits unrestricted use, distribution, reproduction and adaptation in any medium and for any purpose provided that it is properly attributed. For attribution, the original author(s), title, publication source (PeerJ) and either DOI or URL of the article must be cited.
License URL: https://creativecommons.org/licenses/by/4.0/

Keywords: Neuropeptides, GPCRs, Expression, Portunus trituberculatus

Funding: National Natural Science Foundation of China 41776165 31802265 31902350 Natural Science Foundation of Zhejiang Province LY20C190004 KC Wong Magna Fund in Ningbo University This work was supported by the National Natural Science Foundation of China (Nos. 41776165, 31802265 and 31902350), the Natural Science Foundation of Zhejiang Province (LY20C190004) and the KC Wong Magna Fund in Ningbo University. The funders had no role in study design, data collection and analysis, decision to publish, or preparation of the manuscript.

==============================
Neuropeptides and their G protein-coupled receptors (GPCRs) regulate multiple physiological processes. Currently, little is known about the identity of native neuropeptides and their receptors in Portunus trituberculatus. This study employed RNA-sequencing and reverse transcription-polymerase chain reaction (RT-PCR) techniques to identify neuropeptides and their receptors that might be involved in regulation of reproductive processes of P. trituberculatus. In the central nervous system transcriptome data, 47 neuropeptide transcripts were identified. In further analyses, the tissue expression profile of 32 putative neuropeptide-encoding transcripts was estimated. Results showed that the 32 transcripts were expressed in the central nervous system and 23 of them were expressed in the ovary. A total of 47 GPCR-encoding transcripts belonging to two classes were identified, including 39 encoding GPCR-A family and eight encoding GPCR-B family. In addition, we assessed the tissue expression profile of 33 GPCRs (27 GPCR-As and six GPCR-Bs) transcripts. These GPCRs were found to be widely expressed in different tissues. Similar to the expression profiles of neuropeptides, 20 of these putative GPCR-encoding transcripts were also detected in the ovary. This is the first study to establish the identify of neuropeptides and their GPCRs in P. trituberculatus, and provide information for further investigations into the effect of neuropeptides on the physiology and behavior of decapod crustaceans.

Introduction

Numerous neuropeptides have pleiotropic function molecules that are involved in reproduction, development, stress response, immune response, and metabolism (Buratti et al., 2013; Chen et al., 2018; Liu et al., 2019b; Xu et al., 2020; Yang et al., 2014). Several neuropeptides play an essential role in controlling ovarian maturation. These neuropeptides include molt-inhibiting hormone (MIH) (Luo et al., 2015), short neuropeptide F (sNPF) (Bao et al., 2018a), red pigment concentrating hormone (RPCH) (Zeng et al., 2016) and neuropeptide F(NPF) (Tinikul et al., 2017). Notably, MIH and NPF may be promising candidates to induce spawning in Litopenaeus vannamei (González et al., 2010) and Macrobrachium rosenbergii (Tinikul et al., 2017), respectively. Although the draft genome of P. trituberculatus is available, the molecular mechanisms of neuropeptides that regulate reproduction are unknown (Tang et al., 2020).

Neuropeptides predominantly bind to G protein-coupled receptors (GPCRs) to exhibit their functions (Sharabi et al., 2016). GPCRs are the largest gene family of transducing cell surface proteins. The family consists of an extracellular N-terminus, a region of seven transmembrane (7-TM) domains, and an intracellular C- terminus (Davies et al., 2007). The arthropod GPCR gene family is subdivided into four classes, including rhodopsin-like receptors (class A), secretin-like receptors (class B), metabotropicglutamate-like receptors (class C), and atypical GPCRs based on sequence motifs (Brody & Cravchik, 2000). It is challenging to identify GPCRs in related species because these sequences generally do not share as high an overall sequence homology as neuropeptides do (Caers et al., 2012). In decapod crustaceans, several efforts had been made to identify neuropeptide GPCR genes; however, only eight species of GPCRs, including Scylla paramamosain (Bao et al., 2018b), Sagmariasus verreauxi (Buckley et al., 2016), Panulirus argus (Christie, 2020), Carcinus maenas (Oliphant et al., 2018), Paranephrops zealandicus (Oliphant et al., 2020), Macrobrachium nipponense (Sun et al., 2020), Penaeus monodon (Viet Nguyen et al., 2020) and Nephrops norvegicus (Nguyen et al., 2018), have been reported in previous studies or deposited in public databases. Many putative neuropeptide receptors have been tentatively identified on the basis of homology, but only a handful have been functionally deorphaned in crustaceans (Alexander et al., 2018; Bao et al., 2018a; Bao et al., 2018b). Therefore, it is necessary to identify GPCRs to explore the function of neuropeptides in P. trituberculatus.

Advances in Next Generation Sequencing (NGS) have resulted in identification of neuropeptide repertoire of several decapod species, such as S. paramamosain (Bao et al., 2015), Cherax quadricarinatus (Nguyen et al., 2016), N. norvegicus (Nguyen et al., 2018), C. maenas (Oliphant et al., 2018), Procambarus clarkia (Veenstra, 2015), and M. nipponense (Sun et al., 2020), based on transcriptome databases. However, currently, no information is available for identification of neuropeptides and GPCRs in P. trituberculatus. Therefore, identification of neuropeptides and GPCRs represents an important step to unravel the roles of these molecules involved in reproduction.

The swimming crab, Portunus trituberculatus (Crustacea: Decapoda: Brachyura) is common edible crab. It is widely distributed in the estuary and coastal waters of Korea, Japan, China, and Southeast Asia (Meng et al., 2020). The only neuropeptides reported in this species include CHH (Xie et al., 2014), MIH (Wang et al., 2013), and CCAP (Sun et al., 2019). In addition, no studies have reported the P. trituberculatus neuropeptidome. In this study, a total of 47 transcripts encoding putative neuropeptides and 47 predicted GPCR-encoding transcripts in P. trituberculatus were identified based on the CNS transcriptome database. Potential neuropeptides and GPCRs were screened for tissue spatial expression by Reverse transcription-polymerase chain reaction (RT-PCR) and showed that their roles might be involved in regulation of reproduction in P. trituberculatus. The findings of this study provide insights into neuropeptide/GPCR signaling pathways that regulate reproduction in P. trituberculatus.

Materials and Methods

Animals

Crab study protocol was approved by the Animal Ethics Committee of Ningbo University. Healthy swimming crabs (female: 300–350 g; male: 200–250 g) were purchased from the local fisheries market in Zhenhai District, Ningbo City, Zhejiang Province, China. All crabs are in the intermolt stage by observing their morphological characteristics as previously described (Shen et al., 2011). Three female and two male crabs were randomly selected and anesthetized on ice for 20 min before dissection. The cerebral ganglia, eyestalk ganglia and ventral ganglia were then removed surgically, frozen immediately in liquid nitrogen, and stored at −80 °C for further use.

Sample preparation and transcriptome sequencing

Five individual samples from each crab were pooled for extraction of total RNA using Trizol Reagent (Takara) following the manufacturer’s instructions. Genomic DNA was removed using DNase I (Takara). Integrity and purity of RNA were determined using Agilent Bioanalyzer 2100 system (Agilent Technologies, Santa Clara, CA, USA) and the NanoPhotometer® spectrophotometer (IMPLEN, CA, USA), respectively. High-quality RNA samples (OD260/280 = 1.8–2.2, OD260/230 > 2.0) were used for construction of sequencing library. Sequencing libraries were generated using NEBNext® Ultra™ RNA Library Prep Kit for Illumina® (NEB, USA), following the manufacturer’s recommendations. Index codes were used to attribute sequences to each sample. cDNA libraries were then sequenced on an Illumina Hiseq 4000 platform (Illumina, USA) and paired-end reads were generated. Raw reads were processed through a Perl program for quality control. Clean data were obtained by removing reads containing adapter, reads containing ploy-N and low-quality reads from raw data. In addition, Q20, Q30, GC-content and sequence duplication level of the clean data were calculated. Transcriptome assembly was performed usingTrinity with min_kmer_cov set to 2 and all other parameters set default (Grabherr et al., 2011).

Bioinformatic Analysis of neuropeptides and their receptors

Peptides and their receptors were obtained from swimming crab, one transcriptome was assembled from all the ganglia combined. A well-established workflow was used to identify peptides (Bao et al., 2015). NCBI BLAST was used to re-validate the sequences. The nucleotide sequences were then converted to amino acids using Expasy translate tool available online (https://web.expasy.org/translate/) and open reading frames (ORFs) were obtained using ORF finder webserver (https://www.ncbi.nlm.nih.gov/orffinder). All predicted precursors were assessed for the presence of a signal peptide using SignalP5.0 webserver (http://www.cbs.dtu.dk/services/SignalP/). Cysteine to cysteine disulfide bridges were predicted using a webserver (http://disulfind.disi.unitn.it/). Structures of mature peptides were predicted using a well-established workflow (Christie, 2014; Christie & Chi, 2015) and prohormone cleavage sites prediction based on the standards defined by Veenstra (Veenstra, 2000). Other post-translational modifications, including cyclization of N-terminal glutamine or glutamic acid residues and C-terminal amidation at glycine residues were predicted through homologous comparison to known arthropod peptides (Bao et al., 2020; Bao et al., 2015; Christie, 2014; Christie, 2016a; Christie, 2016b; Christie, 2020; Christie & Chi, 2015; Christie et al., 2017; Johnson et al., 2015; Nguyen et al., 2018; Vaudry et al., 2014; Veenstra, 2015; Veenstra, 2016; Wegener et al., 2015; Xu et al., 2015). Illustrator for Biological Sequences (IBS) software was used to generate schematic diagrams of protein domain structures (Ren et al., 2009). Multiple sequence alignments were done using ClustalX. Multiple alignment files were imported to Jalview 2.10.5 software to identify conserved sequence motifs and the LOGO.

GPCR transcript sequences obtained from CNS transcriptomes of P. trituberculatus, were translated using the translate tool in ExPASy and assessed for completeness (predicted protein sequences <150 amino acids in length were deemed too short for accurate vetting and were not reported in this article) (Christie & Yu, 2019). The protein sequences were revalidated by BLAST. GPCR sequences were identified using GPCRHMM webserver (https://gpcrhmm.sbc.su.se/) and only sequences containing seven transmembrance domains (7TM) were retained (Bao et al., 2018b). To further annotate the P. trituberculatus GPCRs for phylogenetic analysis, 7TM domains of all GPCRs (those from this study and previously characterized receptors) were extracted and compiled into one list (Supplemental Information 1). Phylogenetic trees were constructed using SeaView software (Gouy, Guindon & Gascuel, 2009) (PhyML algorithm) and visualized with Figtree v1.4.3 and Adobe Photoshop CS (version 6.0).

Tissue expression of neuropeptides and determination of GPCRs of interest by RT-PCR

To explore tissue expression of neuropeptide transcripts and their GPCR transcripts, nine tissues, including cerebral ganglia, eyestalk ganglia, gill, hepatopancreas, heart, muscle, ovary, ventral ganglia, and Y-organ were harvested from wild female P. trituberculatus (n = 3). All crabs are in ovarian development stage III based on the gonadosomatic index (GSI%, ovary weight ×100/body weight) and external features of the ovary (Wu et al., 2007). Total RNA from tissues was extracted using RNA-Solv Reagent (Omega Biotek, USA), following the manufacturer’s protocol. Total RNA (∼1 µg RNA per tissue) was used for cDNA synthesis using HiFiScript gDNA Removal cDNA Synthesis Kit (CWBIO, China). A total of 32 neuropeptide transcripts, 33 GPCR transcripts, and one housekeeping gene were chosen for RT-PCR. Gene-specific primers were designed using primer 5.0 software based on the CDS of transcripts. Primers used in this article are presented in Supplemental Information 2. RT-PCRs amplifications were carried out using a touch-down PCR to allow maximum PCR products to be amplified with minimal non-specific signal., PCR settings were as follows: 95 °C for 3 min, followed by 35 cycles of touch down, 95 °C for 30 s, 62–57 °C for 30 s (with 1 °C decrease for each of the first 6 cycles) and 72 °C for 20 s. PCR products were separated on 2.0% agarose gel electrophoresis and visualized using GelRed (Biotium).

Results and Discussion

Identification of neuropeptide transcript and peptide prediction

Cerebral ganglia, eyestalk ganglia, and ventral ganglia, each consisting of two males and three females, were used for transcriptome sequencing to identify neuropeptides in P. trituberculatus. All raw reads were submitted to the NCBI SRA under accession: BioProject number PRJNA707143 (SRR13870345, SRR13870346 and SRR13870347). Quality, de novo assembly statistics and annotation summary data are shown in Supplemental Information 3. A total size of 124,237,692 bp of the assembled transcripts were obtained with an average size of 1,784 bp and half of the contigs N50 were at least 3,185 bp long. The shortest and longest assembled sequences were 301 and 28,583 bp, respectively. A total of 47 predicted neuropeptide transcripts and 47 peptide GPCR transcripts were identified using bioinformatics tools, containing 39 A-family GPCRs (Pt-GPCR-A) and 8 B-family GPCRs (Pt-GPCR-B), from central nervous system (CNS, including cerebral ganglia, eyestalk ganglia and ventral ganglia) of P. trituberculatus. Most of the neuropeptides were previously identified in other crustacean and/or insect species (Table 1). Predicted peptides included: adipokinetic hormone corazonin-like peptide (ACP), agatoxin-like peptide (ALP), allatostatin A (AST-A), allatostatinB (AST-B), allatostatin C (AST-C), allatostatin CCC (AST-CCC), bursicon, crustacean cardioactive peptide (CCAP), crustacean hyperglycemic hormone (CHH)/molt-inhibiting hormone (MIH), CCHamide (CCH), crustacean female sex hormone (CFSH), corazonin (Crz), CNMamide (CNM), CRF-like DH44, calcitonin-like diuretic hormone31 (DH31), Carcikinin/ecdysis triggering hormone (ETH), elevenin, glycoprotein-A2 (GPA2), glycoprotein-B5 (GPB5), GSEFLamide, HIGSLYRamide, insulin-like peptide (ILP), kinin, myosuppressin, neuroparsin (NP), neuropeptide F (NPF), short neuropeptide F (sNPF), vasopressin, orcokinin, pigment-dispersing hormone (PDH), proctolin, pyrokinin, red pigment-concentrating hormone (RPCH), RYamide, SIFamide, sulfakinin (SK), tachykinin (TK), Trissin, and Natalisin. Most of these neuropeptides are present in related crustaceans, such as S. paramamosain (Bao et al., 2015), S. olivacea (Christie, 2016b) and C. maenas (Oliphant et al., 2018). Sequence alignments between the precursors from these species showed that all peptides, especially the mature peptides, are conserved. For instance, the predicted mature peptides of Carcikinin (DAGHFFAETPKHLPRIamide), CCAP (PFCNAFTGCamide), and AST-C (pQIRYHQCYFNPISCF) were highly conserved. This suggests that prediction of the putative peptide sequences in this study are mostly accurate.

Table 1 Putative neuropeptide precursors predicted in the central nervous-system transcriptome of P. trituberculatus.

Peptide families	Transcriptome size (bp)	ORF size
(aa)	BLAST matches species	E-value	Ident	Accession number	
ACP
agatoxin-like
allatostatinA
allatostatinB
allatostatinC
allatostatinCCC
bursicon-α
bursicon-β
CCAP
CHH1
CHH2
MIH
CCHamide1
CCHamide2
CFSH
corazonin
CNMamide
CRF-like_DH44
DH31
ETH
elevenin
GPA2
GPB5
GSEFLamide
HIGSLYRamide
ILP
kinin
myosuppressin
neuroparsin
neuroparsin2
neuroparsin3
NPF1
NPF2
sNPF
vasopressin
orcokinin
PDH1
PDH2
proctolin
pyrokinin
RPCH
RYamide
SIFamide
Sulfakinin
tachykinin
trissin
natalisin	722
892
3195
1826
1178
1782
2624
557
660
1245
2190
7048
3469
784
1466
905
1348
1693
449
1373
1803
1436
2007
1388
2811
757
2173
806
2108
814
1494
1454
1080
1498
838
1175
433
520
920
1598
877
880
1171
839
915
1341
1521	101
113
659
314
149
110
148
141
141
139
135
113
189
126
C
117
193
321
111
136
147
120
228
358
758
167
336
100
102
102
110
102
131
126
158
181
79
78
94
338
110
134
78
165
226
201
375	Scylla olivacea
Penaeus vannamei
Panulirus interruptus
Portunus trituberculatus
Scylla paramamosain
Scylla paramamosain
Callinectes sapidus
Callinectes sapidus
Portunus trituberculatus
Portunus trituberculatus
Portunus trituberculatus
Portunus trituberculatus
Cherax quadricarinatus
Nephrops norvegicus
Callinectes sapidus
Carcinus maenas
Scylla olivacea
Cherax quadricarinatus
Nephrops norvegicus
Scylla olivacea
Carcinus maenas
Cherax quadricarinatus
Cherax quadricarinatus
Homarus americanus
Cherax quadricarinatus
Nephrops norvegicus
Scylla paramamosain
Scylla paramamosain
Portunustrituberculatus
Scylla paramamosain
Nephrops norvegicus
Scylla paramamosain
Scylla paramamosain
Scylla paramamosain
Portunus pelagicus
Cancer borealis
Nephrops norvegicus
Scylla paramamosain
Nephrops norvegicus
Scylla paramamosain
Portunus trituberculatus
Nephrops norvegicus
Scylla paramamosain
Scylla paramamosain
Scylla paramamosain
Cherax quadricarinatus
Scylla paramamosain	3.15E−52
1.00E−42
0.00E+00
0.00E+00
9.00E−81
1.00E−74
6.00E−102
4.00E−79
4.00E−100
5.00E−100
1.00E−91
3.00E−78
1.00E−11
8.00E−20
2.00E−155
3.00E−48
2.00E−117
1.00E−26
8.00E−15
4.06E−72
1.40E−75
3.00E−75
2.00E−79
3.00E−146
3.00E−51
1.00E−26
8.00E−174
1.00E−67
2.00E−67
2.00E−51
2.00E−12
3.00E−64
3.00E−62
2.00E−86
2.00E−106
3.00E−116
1.00E−25
4.00E−27
1.00E−22
0.00E+00
3.00E−68
3.00E−50
1.00E−47
2.00E−40
6.00E−103
1.00E−31
0.00E+00	95.29%
71.68%
52.87%
100%
91.28%
100%
97.30%
86.62%
100%
100%
100%
100%
61.40%
45.54%
93.72%
78.30%
97.93%
77.94%
68.89%
94.11%
85.71%
87.50%
86.51%
68.82%
37.08%
42.29%
87.78%
100%
100%
95.88%
44.87%
96.08%
84.73%
96.83%
98.10%
93.92%
63.29%
98.72%
64.86%
85.67%
100%
63.04%
98.72%
84.25%
82.53%
44.67%
86.70%	Christie (2016a), Christie (2016b)
XP027234917.1
BAF64528.1
QCI34367.1
ALQ28578.1
ALQ28598.1
ACG50067.1
ACG50066.1
AVK43051.1
ACB46189.1
MPC59687.1
MPC23433.1
AWK57506.1
QBX89034.1
ADO00266.1
AVA26882.1
Christie (2016a), Christie (2016b)
AWK57510.1
QBX89032.1
Christie (2016a), Christie (2016b)
Christie (2016a)
AWK57521.1
AWK57522.1
AYH52118.1
AWK57523.1
QBX89050.1
ALQ28594.1
ALQ28580.1
AVK43050.1
ALQ28589.1
QBX89061.1
ALQ28586.1
ALQ28587.1
ALQ28574.1
AUT12056.1
ABY82345.1
QBX89064.1
ALQ28583.1
QBX89067.1
ALQ28575.1
MPC25086.1
QBX89071.1
ALQ28576.1
ALQ28597.1
ALQ28591.1
AWK57547.1
ALQ28592.1	
Notes.

C, C-terminal partial protein

Figure 1 Identification and molecular characterization of P. trituberculatus ACP (A), ALP (B), AST-A (C), AST-B (D) and AST-C/AST-CCC (E).

Schematic diagrams show structures of neuropeptide precursors, including signal peptide (SP), mature peptide, putative cleavage sites, and related peptide. Predicted mature peptide amino acid sequence of AST-A, AST-B, and AST-C/AST-CCChave been aligned using Clustal X2.1, shown conserved sequence motifs by Jalview 2.10.5.

Adipokinetic hormone–corazonin like peptide (ACP)/Agatoxin-like peptide (ALP)/Allatostatin (AST)/Bursicon and CCAP

A putative ACP precursor was predicted from the de novo transcriptome of P. trituberculatus central nervous system. The predicted encoded complete ACP precursor of 101 amino acids (aa), with 12 aa mature peptides, pQITFSRSWVPQamide (Supplemental Information 4 and Fig. 1A), is a highly conserved decapod ACP peptide (Christie, 2016a; Christie, 2016b; Christie & Chi, 2015). One Pt-ALP transcript was identified in the current de novo transcriptome assembly of P. trituberculatus. The transcript encodes for 113 aa protein, with 21aa signal peptide. PtALP has two cleavage sites (K66K and K113), which if processed results in cleavage of the mature peptide WRSCIRRGGACDHRPNDCCYNSSCRCNLWGTNCRCQRMGIFQQWamide. It has 8 cysteine residues that presumably allow for the formation of four identically positioned disulfide bridges (Supplemental Information 4 and Fig. 1B), and amidated C-terminus which are characteristic of decapod species (Christie et al., 2020). Based on their distinct structural differences, allatostatins are classified into three different families (A-type, B-type and C-type) (Stay & Tobe, 2007; Verlinden et al., 2015) In crustaceans, the existence of AST-As, AST-Bs and AST-Cs were first identified from Cancer borealia (Skiebe & Schneider, 1994), Cancer productus (Fu et al., 2005), Homarus americanus (Ma et al., 2009b), respectively. A total of 4 transcripts were detected that encoded for the precursor of AST family including, AST-A, AST-B, AST-C, and AST-CCC. The precursor AST-A had a region of the predicted signal peptide (27 aa), followed by 35 predicted peptides separated by multiple dibasic cleavage sites. The conserved motif recorded was XYXFGLamide (Supplemental Information 4 and Fig. 1C), such as AGPYSFGLamide sequence, which is common insects and other crustaceans in the AST-A family (Bao et al., 2015; Christie, Stemmler & Dickinson, 2010; Stay & Tobe, 2007). A single transcript was observed in the transcriptome assembly. The putative neuropeptide precursor encoded 314 aa and has a 19 aa signal peptide. This precursor had 11 predicted mature peptides, containing the conserved XWXXXXGXWamide motif (Supplemental Information 4 and Fig. 1D), such as AGWSSMRGAWamide sequence, which is a signature motif in AST-B family (Christie, Stemmler & Dickinson, 2010). The predicted sequence of the AST-C precursor comprised 149 aa with a predicted signal peptide of 21 aa. The predicted mature peptide had15aa, with a highly conserved PISCF motif with pQIRYHQCYFNPISCF (a disulfide bridge between two cysteine residues) sequence (Bao et al., 2020; Bao et al., 2015; Ma et al., 2009b; Nguyen et al., 2016). AST-CCC precursor showed a 110aa complete sequence, 24aa single peptide and 15aa mature peptide (Supplemental Information 4 and Fig. 1E). An AST-C-like peptide precursor, was also detected in P. trituberculatus. The AST-CCC shares a conserved motif (SYWKQCAFNAVSCFamide) with the AST-C sequence (Supplemental Information 4 and Fig. 1E). In crustaceans, the first isoform of burscion was identified from C. maenas (Ma et al., 2009a; Wilcockson & Webster, 2008). In the transcriptome assembly, two transcripts, which encoded bursicon alpha and bursicon beta, precursor of 148 aa and 141 aa, respectively, were obtained. The two precursors had a signal peptide, immediately followed by adjacent mature peptide with 11 conserved cysteine residues (Supplemental Information 4 and Fig. 2A). The structures of the mature peptides were highly conserved across other decapod crustaceans, such as Callinectes sapidus and P. argus (Christie, 2020). CCAP was originally identified from C. maenas via reverse phase high-performance liquid chromatography (RP-HPLC) procedure (Stangier et al., 1987) and the first CCAP transcript was predicted in Daphnia pulex via transcriptomics (Gard et al., 2009). In this study, One transcript was predicted to encode CCAP precursor of 141 aa, containing a predicted signal peptide of 30 aa and two cleavage sites (K47R and K59K). The peptide released a mature peptide with PFCNAFTGCamide (a disulfide bridge between two cysteine residues) motif sequence (Supplemental Information 4 and Fig. 2B).

Figure 2 Identification and molecular characterization of P. trituberculatus burscionα/burscionβ (A), CCAP (B), CHH1/CHH2 (C), and MIH (D).

Schematic diagrams show structure of neuropeptide precursors, including signal peptide (SP), mature peptide, putative cleavage sites and related peptide. Comparative mature peptide sequence alignment of burscionα and burscionβ for P. trituberculatus with C. sapidus and P. argus respectively. Predicted mature peptide amino acid sequence of CHH1/CHH2 has been aligned using ClustalX2.1, shown conserved sequence motifs by Jalview 2.10.5.

Crustacean hyperglycemic hormone (CHH)/Molt-inhibiting hormone (MIH)/Crustacean female sex hormone (CFSH)/CCHamide (CCH) and Corazonin (Crz)

CHH was first purified and sequenced from eyestalks of C. maenas (Kegel et al., 1989). In 2006, CHH splice variants have confirmed in C. sapidus (Choi, Zheng & Watson, 2006). Two transcripts were identified to putatively encode complete CHH precursors with 139 aa and 135 aa. Pt-CHH1 and Pt-CHH2 are products of alternative splicing. All two complete sequences consist of 26 aa signal peptide, a CHH-precursor-related peptide (CPRP) and CHH mature peptides with 71 aa and 73 aa, respectively (Supplemental Information 4 and Fig. 2C). The two mature peptides had six cysteine residues, and the six cysteines were aligned (Fig. 2C). Characteristics of these peptides were similar as we previously studied (Xie et al., 2014). The first putative MIH was isolated and sequenced in C. maenas (Klein et al., 1993). The predicted sequence of the MIH precursor comprised 113 aa, with 35 aa signal peptide. The sequence lacked CHH-PRP and had an additional conserved glycine in position 12 of the putative mature peptide (Gly12) (Supplemental Information 4 and Fig. 2D). In addition, the mature peptide had six cysteine residues and could form three disulfide bridges, which is a characteristic of CHH family. Two transcripts of CCH precursors were identified in the de novo transcriptome assembly of P. trituberculatus. The two transcripts encode for 189 aa and 126 aa, each with a signal peptide (Supplemental Information 4 and Fig. 3A). The conserved motif for Pt-CCHamide mature peptide was XCXXY/FGHSCXGAHamide (a disulfide bridge between two cysteine residues), which is conserved in other decapod species, such as C. quadricarinatus (Nguyen et al., 2016) and P. clarkia (Veenstra, 2015). CFSH was initially isolated from sinus glang (SG) of C. sapidus via RP-HPLC chromatograms and subsequently cDNA was cloned in this species (Zmora & Chung, 2014). A putative CFSH transcript with partial C-terminus of 223 aa was predicted from the central nervous system of P. trituberculatus. The precursor CFSH had a region of the predicted signal peptide (22 aa) and a K55R cleavage site, followed immediately by the mature peptide of 167 aa (Supplemental Information 4 and Fig. 3B). The mature peptide had eight cysteine residues, which were conserved in other decapod CFSHs (Veenstra, 2016). The first Crz sequence was identified in C. borealis via mass spectrometry (Li et al., 2003). Also, Crz precursor have been identified via transcriptome analysis from L. vannamei (Ma et al., 2010) and Daphnia carinata (Christie et al., 2010). A single transcript was identified to putatively encode complete Crz precursor with 117 aa. Crz precursor had a 19 aa signal peptide, followed immediately by the mature peptide, and a R32KR cleavage site (Supplemental Information 4 and Fig. 3C). The conserved motif sequence of Crz mature peptide was pQTFQYSRGWTNamide, a highly conserved Crz motif in other crustaceans and insects (Alexander et al., 2018; Nguyen et al., 2016).

Figure 3 Identification and molecular characterization of P. trituberculatus CCH1/CCH2 (A), CFSH (B), Crz (C), CNM (D), CRF-like DH44 (E), and DH31 (F).

Schematic diagrams show structure of neuropeptide precursors, including signal peptide. (SP), mature peptide, putative cleavage sites, and related peptide. Predicted mature peptide amino acid sequence of CCH1 and CCH2 have been aligned using Clustal X2.1, shown conserved sequence motifs by Jalview 2.10.5.

CNMamide(CNM)/CRF-like DH44/Calcitonin-like diuretic hormone 31(DH31)/ Carcikinin/ecdysis triggering hormone(ETH) and Elevenin

A CNMamide transcript was present in the assembly of P. trituberculatus. The precursor putatively comprised 193 aa with a 30 aa predicted signal peptide. A 10 aa C-terminal amidated CNMamide mature peptide is released after cleavage of two dibasic cleavage sites (Supplemental Information 4 and Fig. 3D). The predicted mature peptide, VMCHFKICNMamide (a disulfide bridge between two cysteine residues) is conserved in other crustacean species (Veenstra, 2015; Veenstra, 2016). A single transcript was identified which encoded a putative CRF-like DH44 precursor. This transcript encodes a 321 aa full-length protein with a 19 aa predicted signal peptide (Supplemental Information 4 and Fig. 3E). This precursor has a 45 aa mature peptide, with the sequence, NSGLSLSIDASMKVLREALYLEIARKKQRQQLQRAQHNKALLNTIamide, which is conserved in other decapod species, such as H. americanus (Christie et al., 2017) and M. rosenbergii (Wegener et al., 2015). The first DH31 has been identified from D. pulex via transcriptomics (Gard et al., 2009). The predicted sequence of the DH31 precursor comprised 111 aa, with a predicted signal peptide of 20 aa at the N-terminal, two DH31-precursor-related peptides and, dibasic K62R and K98R cleavage sites, that if processed results in release of a DH31 mature peptide (Supplemental Information 4 and Fig. 3F). Pt-DH31 mature peptide is homologous to N. norvegicus DH31 (Christie et al., 2009; Nguyen et al., 2018). Carcikinin has previously been identified as a crustacean orthologue of insect ecdysis triggering hormone (ETH) (Veenstra, 2016). The first carcikinin has been predicted from D. pulex via transcriptomics (Gard et al., 2009). A complete carcikinin precursor comprised 136aa, with a predicted signal peptide of 19aa, followed immediately by a 16aa mature peptide. The predicted mature peptide had conserved DAGHFFAETPKHLPRIamide motif (Supplemental Information 4 and Fig. 4A), which is highly conserved in other decapod species, such as S. paramamosain and M. rosenbergii (Minh Nhut et al., 2020). The putative transcript encoded a complete elevenin precursor with 147 aa. The precursor had a 27 aa predicted signal peptide, followed by a 20 aa mature peptide, and contained the conserved XXXXDCR-K/R-FVFAPXCRGIIA motif (a disulfide bridge between two cysteine residues) (Supplemental Information 4 and Fig. 4B), present in other crustaceans (Christie et al., 2017; Nguyen et al., 2016).

Figure 4 Identification and molecular characterization of P. trituberculatus ETH (A), Elevenin (B), GPA2 (C), GPB5 (D), GSEFLamide (E), and HIGSLYRamide (F).

Schematic diagrams show structure of neuropeptide precursors, including signal peptide (SP), mature peptide, putative cleavage sites, and related peptide. Comparative mature peptide sequence alignment of elevenin for P. trituberculatus with C. quadricarinatus and H. americanus. Predicted mature peptide amino acid sequence of GSEFLamide and HIGSLYRamide have been aligned using ClustalX2.1, shown conserved sequence motifs by Jalview 2.10.5.

Glycoprotein-A2 (GPA2)/Glycoprotein-B5 (GPB5)/GSEFLamide (GSE)/HIGSLYRamide and Insulin-like peptide (ILP)

A single transcript was shown to encode GPA2 precursor with 120 aa. The predicted mature peptide of 102 aa was encoded immediately after the predicted signal peptide of 18 aa (Supplemental Information 4 and Fig. 4C). These peptides share highly conserved sequence with other described decapod GPA2 isoforms, for example with 87.5% identity with amino acid sequence to C. quadricarinatus GPA2 isofrom (Nguyen et al., 2016). Pt-GPA2 mature peptide had 10 cysteine residues, possibly forming 5 disulfide bridges. Analysis of the P. trituberculatus GPA2 by DiANNA showed disulfide bridging between the first and tenth, second and eighth, third and fifth, fourth and ninth, and sixth and seventh cysteine residues of the peptide. The peptide sequence was homologous to previously reported decapod GPA2 isoforms (Christie et al., 2017; Christie & Yu, 2019). The nucleotide sequence obtained from the transcriptome data had a 2007 bp putative sequence coding for a complete GPB5 precursor with 228 aa (Supplemental Information 4 and Fig. 4D). The precursor had no predicted signal peptide and the mature peptide had 10 cysteine residues, possibly forming 5 disulfide bridges. Analysis of this GPB5 isoform using DiANNA showed disulfide bridging between its first and tenth, second and fifth, third and fourth, sixth and eighth, and seventh and ninth cysteine residues. The isoform showed high sequence similarity to known decapod GPB5 isoforms, for example 86.51% identity in amino acid sequence to C. quadricarinatus GPAB5 isoform (Nguyen et al., 2016). One transcript was identified that encoded a putative GSEFLamide precursor. The precursor encodes for 358 aa, containing a 25 aa predicted signal peptide followed by 18 mature peptides flanked by multiple cleavage sites (Supplemental Information 4 and Fig. 4E). The 18 predicted mature peptides were 7 to 8 aa in length and each had a conserved XGSEFLamide motif. The first putative transcript encoding HIGSLYRamide has been identified in C. maenas via in silico analyses (Christie et al., 2008). A partial precursor sequence for HIGSLYRamide was identified, comprising 758 aa and had no signal peptide (Supplemental Information 4 and Fig. 4F). Multiple dibasic KR/RR cleavage sites indicate presence of 18 mature peptides of 8 aa in length. The conserved motif of these mature peptide was H-I/L-GSL-L/Y-Ramide. A single transcript was identified to encode ILP precursor with 167 aa. The precursor had a 21 aa predicted signal peptide and 2 conserved regions(21 aa A chain and 36 aa B chain) that formed the mature hormone (Supplemental Information 4 and Fig. 5A). Pt-ILP mature peptide had 6 cysteine residues located in B chain and A chain, possibly forming 3 disulfide bridges. Analysis showed highly conserved sequence features of arthropod insulin-like peptide (Veenstra, 2020).

Figure 5 Identification and molecular characterization of P. trituberculatus ILP (A), Kinin (B), Myosuppressin. (C), NP1/NP2/NP3 (D), NPF1/NPF2 (E), and sNPF (F).

Schematic diagrams show structure of neuropeptide precursors, including signal peptide (SP), mature peptide, putative cleavage sites, and related peptide. Predicted mature peptide amino acid sequence of Kinin, NP1/NP2/NP3, NPF1/NPF2, and sNPF have been aligned using Clustal X2.1, shown conserved sequence motifs by Jalview 2.10.5.

Kinin/Myosuppressin/Neuroparsin (NP)/Neuropeptide F (NPF) and Short neuropeptide F (sNPF)

In crustaceans, the first kinin peptide was identified from Penaeus vannamei via mass spectrometry (Nieto et al., 1998). The predicted sequence of the kinin precursor was identified in the de novo assembly with no signal peptide. The precursor encodes 336 aa and has 12 predicted mature peptides, separated by multiple dibasic cleavage sites (Supplemental Information 4 and Fig. 5B). In addition to the mature peptide of 13 aa, SGDSKHGRFSAWAamide, another 11 mature peptides with 6 to 7 aa in length were identified. These mature peptides had conserved X-F/L-N/S-X-WAamide, e.g., QAFSAWAamide motifs, including QALNVWAamide, which are signature motifs in kinin family (Christie, Stemmler & Dickinson, 2010). In decapod crustaceans, the first myosuppressin has been identified via mass spectrometry (Stemmler et al., 2007). A complete myosuppressin precursor comprised 100 aa with a predicted signal peptide of 24 aa. The precursor had two cleavage sites (K84R and R97), that are processed to release the mature peptide with pQDLDHVFLRFamide sequence (Supplemental Information 4 and Fig. 5C). Pt-myosuppressin had the same motif which is highly conserved in other known crustacean species (Christie, Stemmler & Dickinson, 2010). In crustaceans, the first putative NP encoding transcripts have been predicted in C. maenas via transcripomics (Ma et al., 2009a). Three transcripts were shown to putatively encode complete NP precursors with 102 to 110 aa. All of the precursors had a predicted signal peptide of 25 to 28 aa, followed immediately by 74 to 85 aa mature peptides (Supplemental Information 4 and Fig. 5D). These mature peptides had 12 conserved cysteine residues, possibly forming 6 disulfide bridges, one of which is similar to peptides reported previously in other decapod species (Ma et al., 2010; Veenstra, 2016).In crustaceans, the first putative transcripts encoding NPF have been predicted in C. maenas via transcriptomics (Christie et al., 2008). Two transcripts were identified to encode 102 aa (NPF1) and 131 aa (NPF2) peptide precursors. The two precursors had predicted signal peptide of 20 aa and 26 aa, respectively (Supplemental Information 4 and Fig. 5E).The predicted 67 aa (NPF1) and 70 aa (NPF2) mature peptide are encoded immediately after the signal peptide and both have the conserved RPRFamide motif, a signature motif in NPF family (Christie, Stemmler & Dickinson, 2010). The first sNPF was identified from M. rosenbergii (Sithigorngul et al., 1998). A transcript of sNPF precursor was identified from the transcriptome data. The precursor encodes 126 aa, containing a 25 aa predicted signal peptide, followed by three dibasic cleavage sites that when processed results in release of three mature sNPF peptides (Supplemental Information 4 and Fig. 5F). The mature peptides were 9 to 12 aa in length and had conserved PXXRLRFamide motifs, including KDARTPALRLRFamide and APPSMRLRFamide which are signature motifs of sNPF family (Christie, Stemmler & Dickinson, 2010).

Figure 6 Identification and molecular characterization of P. trituberculatus Vasopressin (A), Orcokinin (B), PDH-1/PDH-2 (C), Proctolin (D), Pyrokinin (E), and RPCH (F).

Schematic diagrams show structure of neuropeptide precursors, including signal peptide (SP), mature peptide, putative cleavage sites, and related peptide. Predicted mature peptide amino acid sequence of orcokinin, PDH-1/PDH-2, and pyrokinin have been aligned using Clustal X2.1, shown conserved sequence motifs by Jalview 2.10.5.

Vasopressin/Orcokinin/Pigment-dispersing hormone (PDH)/Proctolin and Pyrokinin

The vasopressin was first predicted from D. pulex (Stafflinger et al., 2008). A single transcript encoded a complete vasopressin-neurophysin precursor with 158 aa. The precursor comprised a 20 aa signal peptide and a dibasic cleavage site at K31R that, if processed, is predicted to release a vasopressin mature peptide and a neurophysin mature peptide (Supplemental Information 4 and Fig. 6A). Pt-vasopressin mature peptide had a conserved CFITNCPPGamide motif (with a disulfide bridge between two cysteine residues). This motif has been reported in other decapod species, such as Portunus pelagicus (Saetan et al., 2018) and S. paramamosain (Lin, Wei & Ye, 2020). A mature 126 aa peptide with fourteen cysteine residues, probably forming seven disulfide bridges, which is characteristic of neurophysin family was identified (Bao et al., 2020). The first orcokinin peptide has been identified from C. maenas via enzyme immunoassay (Bungart et al., 1995). The putative transcript encoded a complete orcokinin precursor with 181 aa. The precursor had a predicted signal peptide with 21 aa and multiple cleavage sites that potentially results in releasing of 8 mature peptides (Supplemental Information 4 and Fig. 6B). Seven peptides shared a conserved C-terminus: NFDEIDRSXFGFX motif with other known crustacean species orcokinins. Another mature peptide (named orcokinin-1) had a FDAFTTGFGHS motif, which was identified as decapod orcomyotropin isoform (Christie, Stemmler & Dickinson, 2010). In crustaceans, the first PDH has been identified from the eyestalk ganglia of Pandalus borealis (Fernlund, 1976). Two PDH precursors were predicted from the central nervous system transcriptome of P. trituberculatus, including Pt-PDH-1 and Pt-PDH-2. These precursors encode 79 aa (PDH-1) and 78 aa (PDH-2) peptides, respectively (Supplemental Information 4 and Fig. 6C). The two precursors had a 22 aa signal peptide, followed by a dibasic KR and RR cleavage site that is processed to release a mature 18 aa peptide. The two mature peptides exhibited highly conserved sequences reported in PDH of other known crustacean species (Bao et al., 2015; Christie, Stemmler & Dickinson, 2010; Oliphant et al., 2018). The first proctolin-like peptide has been isolated from pericardial organs in Cardisoma carnifex (Sullivan, 1979). Recently, the first putative proctolin-encoding transcript was identified from L. vannamei via transcriptomics (Ma et al., 2010). One proctolin precursor 94 aa in length was identified. The precursor had a 23 aa signal peptide followed immediately by the predicted mature peptide (Supplemental Information 4 and Fig. 6D). Pt-Proctolin had a conserved RYLPT motif, completely conserved in other crustacean species (Christie, Stemmler & Dickinson, 2010; Li et al., 2003). In decapods, pyrokinin has been first isolated and identified from P. vannamei (Torfs et al., 2001). A pyrokinin transcript was identified that encoded a 338 aa precursor. The precursor encoded a predicted 30 aa signal peptide and 10 mature peptides of 7 to 12 aa flanked by multiple dibasic cleavage sites (Supplemental Information 4 and Fig. 6E). These mature peptides had conserved F-A/S-PR-P/Lamide motif, such as FAPRPamide, FSPRLamide, which were similar to the 5aa conserved in the C-terminal region, and FXPRLamide, which is a signature motif in pyrokinin family (Choi & Vander Meer, 2012).

Figure 7 Identification and molecular characterization of P. trituberculatus RYamide (A), SIFamide (B), Sulfakinin (C), Tachykinin (D), Trissin (E), and Natalisin (F).

Schematic diagrams show structure of neuropeptide precursors, including signal peptide (SP), mature peptide, putative cleavage sites, and related peptide. Predicted mature peptide amino acid sequence of RYamide, Sulfakinin, Tachykinin, and natalisin have been aligned using Clustal X2.1, shown conserved sequence motifs by Jalview 2.10.5.

Red pigment-concentrating hormone (RPCH)/RYamide/SIFamide/ Sulfakinin (SK)/Tachykinin (TK)/Trissin and Natalisin

The first RPCH has been purified and chemical structural has also been identified from Leander adspersus (Carlsen & Josefsson, 1976). A complete RPCH precursor was predicted from the central nervous system transcriptome. The precursor encoded 110 aa, containing a predicted 25 aa signal peptide, followed immediately by a 9 aa mature peptide and a dibasic cleavage site at K35R (Supplemental Information 4 and Fig. 6F). The mature peptide had a conversed pQLNFSPGWamide motif, which is highly conserved in RPCH of other crustacean species, such as S. paramamosain (Zeng et al., 2016) and L. vannamei (Chen et al., 2018). The first RYamide has been identified from Cancer borealis via mass spectrometry (Li et al., 2003). To date, nothing is known about the bioactivity of this peptide in crustaceans. A single transcript encoding RYamide precursor was identified from the transcriptome data. The precursor encoded 134 aa, including a predicted 25 aa signal peptide and multiple dibasic cleavage sites that when cleaved results in release of three different mature peptides (Supplemental Information 4 and Fig. 7A). These mature peptides share conserved “RYamide” motif, with the sequence FYSQRYamide, which is a RYamide family signature (Christie, Stemmler & Dickinson, 2010). The first SIFamide has been identified from P. monodon via RP-HPLC (Sithigorngul et al., 2002). One transcript was identified to encode a complete SIFamide precursor with 78 aa. The precursor had a predicted 27 aa signal peptide followed immediately by the mature peptide with a C-terminal dibasic cleavage site (K41R) (Supplemental Information 4 and Fig. 7B). The mature peptide had a conversed GYRKPPFNGSIFamide motif, which showed significant similarity with other known crustacean SIFamide motifs (Christie, Stemmler & Dickinson, 2010). In crustaceans, the first sulfakinins have been isolated and identified from P. monodon (Johnsen et al., 2000). The putative transcript encoded a complete sulfakinin precursor with 165 aa. The precursor had a 31 aa predicted signal peptide, followed by three dibasic cleavage sites that when processed may result in release of two mature sulfakinin peptides (Supplemental Information 4 and Fig. 7C). The mature peptides, included EFDDYGHMRFamide and GSASDDYQDDYGHLRFamide, and had a C-terminal conserved DYGH-M/L-RFamide motif, which is a sulfakinin family signature (Christie, Stemmler & Dickinson, 2010). The first crustacean tachykinin peptide have been identified from C. borealis (Christie et al., 1997). A single transcript was identified that encoded tachykinin precursor. The precursor encoded 226 aa, containing a 20 aa signal peptide, followed by 6 TK mature peptides each 9 aa long, and each mature peptide was flanked by cleavage sites (Supplemental Information 4 and Fig. 7D). All six TK peptides shared the conserved XPSGFLGMRamide motif, a signature motif in other crustacean species TK peptides (Christie, Stemmler & Dickinson, 2010). A complete trissin precursor transcript 201 aa long was identified. The precursor encoded a predicted signal peptide of 24 aa, followed immediately by a 29 aa mature peptide (Supplemental Information 4 and Fig. 7E). The predicted trissin peptide had a conserved STVSCDSCGPECQTACGTKNFRACCFNFL motif with six cysteine residues, possibly forming three disulfide bridges. Disulfide bridges were predicted using DiANNA between the first and sixth, second and fourth, and third and fifth cysteine residues. This characteristic motif of Pt-trissin has been reported previously in other crustacean species, such as H. americanus (Christie et al., 2017), P. clarkia (Veenstra, 2015) and other transcriptomes with trissin annotated (Bao et al., 2020; Veenstra, 2016). The nucleotide sequence produced from the transcriptome data showed a 1521 bp putative coding for a complete WXXXRamide precursor with 375 aa. The precursor comprised a predicted signal peptide of 20 aa and multiple dibasic cleavage sites that results in release of 12 mature WXXXRamide peptides with 8 to 17 aa (Supplemental Information 4 and Fig. 7F). These mature peptides had a conserved WXXXnRamide (where Xn represents a glycine or nothing) motif.

Lost neuropeptide transcripts in the CNS transcriptome of P. trituberculatus

Notably, no transcripts encoding putative allatotropin, EH, and FLRFamide precursors were identified from the P. trituberculatus transcriptome. Allatotropin (AT) neuropeptide is a ubiquitous bioactive molecule that stimulates juvenile hormone production by corpora allata in insects (Bednár et al., 2017; Elekonich & Horodyski, 2003). Currently, no studies have explored on juvenile hormone in crustacean species. In addition, no study has reported AT receptor in P. trituberculatus transcriptome. Lack of identification of an AT signaling system in P. trituberculatus implies that this peptide group has been lost during evolution in decapod crustaceans (Bao et al., 2015; Christie & Pascual, 2016; Christie et al., 2020; Christie et al., 2017; Christie & Yu, 2019; Fu et al., 2005; Nguyen et al., 2018; Veenstra, 2016). EH was originally identified as a brain-derived hormone that is synthesized in ventromedial disposition in the brains of most insects (Horodyski, Riddiford & Truman, 1989). This peptide might play a role in regulation of ecdysis, a common physiological process in insects (Scott et al., 2020). Identification of EH precursors in other members of the Portunidae, such as S. paramamosain (Bao et al., 2015) and S. olivacea (Christie, 2016b), implies that these signaling systems are likely to be present in P. trituberculatus. A putative EH transcript was identified in the testis transcriptome database of P. trituberculatus (Supplemental Information 7). In insects, FLRFamide is reported in endocrine and paracrine cells of the midgut (Kingan et al., 1997), however, only a few decapod crustacean species have been reported to harbor these peptides in CNS (Bao et al., 2015; Liu et al., 2019c). Analysis showed that the peptides were expressed in different tissue among diverse species. Therefore, FLRFamide expression in CNS might not have been similar as that in S. paramamosain. Differences in expressions may explain the absence of this neuropeptide in the transcriptome of P. trituberculatus. Therefore, studies should analyze midgut RNA-seq data to explore expression of this neuropeptide.

Neuropeptide GPCRs prediction

A total of 47 candidate neuropeptide GPCR transcripts were predicted from P. trituberculatus. In order to identify orthologs for these GPCRs, deorphanized neuropeptide GPCRs from crustaceans and insects were collected as main reference sequences. Analysis of the phylogenetic tree of these GPCRs, 39 belonged to the A-family (Fig. 8), and eight belonged to the B-family (Fig. 9). However, the receptors for insulin-like peptides (ILPs) and neuroparsin (NP) are not GPCRs (Brogiolo et al., 2001; Vogel, Brown & Strand, 2015). In addition, the receptors for CFSH and orcokinin have not been currently identified in crustaceans.

Figure 8 Phylogeny of the A-family neuropeptide GPCRs. Cladogram of neuropeptide GPCRs showing connections in the clustermap of A-family neuropeptide receptors.

Pt-GPCR, Portunus trituberculatus GPCR; Sp-GPCR, Scylla paramamosain GPCR; Bm-GPCR, Bombyx mori GPCR; Dp-GPCR, Daphnia pulex GPCR; Dm-GPCR, Drosophila melanogaster GPCR; Nn-GPCR, Nephrops norvegicus GPCR;Nl-GPCR, Nilaparvata lugens GPCR; Pc-GPCR, Procambarus clarkia GPCR; Sv-GPCR, Sagmariasus verreauxi GPCR; Tu-GPCR, Tetranychus urticae GPCR; Zn-GPCR, Zootermopsis nevadensis GPCR; Tc-GPCR, Tribolium castaneum GPCR.

Figure 9 Phylogeny of the B-family neuropeptide GPCRs.

Cladogram of neuropeptide GPCRs showing connections in the clustermap of B-family neuropeptide receptors. Pt-GPCR, Portunus trituberculatus GPCR; Sp-GPCR, Scylla Paramamosain GPCR; Dm-GPCR, Drosophila melanogaster GPCR; Pc-GPCR, Procambarus clarkia GPCR; Sv-GPCR, Sagmariasus verreauxi GPCR; Tu-GPCR, Tetranychus urticae GPCR; Zn-GPCR, Zootermopsis nevadensis GPCR; Mn-GPCR, Macrobrachium nipponense GPCR.

A-family GPCRs

In P. trituberculatus, 39 A-family neuropeptide GPCRs were identified. It is relatively small putative A-family GPCR transcripts in the swimming carb compare with those identified in other related crustaceans (Bao et al., 2015; Johnson et al., 2021). This could be explained as the following reasoning: (1) due to low expression levels and relatively small amounts of RNAseq reads, it is extremely difficult to be identified; (2) different expression in various tissues. It is far from to identify amount of GPCR transcripts only using CNS transcriptome database. More tissues transcriptome are needed to identified more GPCR transcripts; (3) deeper sequencing is required to identify more GPCR transcripts, which have been reported in other Portunidae species. Analysis of the phylogenetic tree of A-family GPCRs showed that 32 Pt-GPCR-A sequences clustered with the known GPCR orthologs (Fig. 8). The class A family comprises AST-AR, AST-BR, AST-CR, Burscion receptor, CCAPR, Corazonin receptor, ETHR, FMRFamide receptor, Moody receptor, proctolin receptor, SIFamide receptor, GPA2/GPB5receptor, CCHamide receptor, NPFR, RYamide receptor, sNPFR, RYamidereceptor, myosuppressin receptor, pyrokinin receptor, natalisin receptor, tachykinin receptor, and Vasopressin receptor. The corresponding neuropeptides of all these receptors were identified, except for FMRFamide receptor and Moody receptor which were present in P. trituberculatus. Similar findings were observed for S. paramamosain (Bao et al., 2015). These two species had FMRFamide receptor and Moody receptor, however their corresponding neuropeptides were not identified. This may be attributed to use of bioinformatics methods to predict neuropeptide GPCR and expression profiles which may not be sufficient. Therefore, the complete ORF sequences of these Pt-GPCRs should be cloned for accurate identification. Notably, a few neuropeptides were not identified in the receptors in this study, such as trissin receptor, RPCH receptor and ACP receptor. This can be attributed the process of filtering out of the sequences resulting in loss of short protein sequences (<150 aa). Therefore, further sequencing or High-Quality Genome Assembly is required to identify trissin receptor, RPCH receptor and ACP receptor in P. trituberculatus. The identification and characterization of the CHH/MIH receptor has remained elusive, but it is assumed that the receptors for the CHH family are GPCRs. In the silk moth Bombyx mori, two putative GPCRs identified BNGR-A2 and A34 as ITP receptors, and BNGR-A24 is an ITP-like receptor (member of CHHR family) (Nagai et al., 2014). Based on this discovery, Veenstra identified three transcripts as putative CHH-like receptors (CHHRs) in P. clarkia (Veenstra, 2015). Subsequently, CHHR orthologs have been identified in more than ten other decapod crustacean species (Buckley et al., 2016; Johnson et al., 2021; Oliphant et al., 2018; Tran et al., 2019). One putative GPCR (Pt-GPCR-A32) was identified as CHH-like receptor, as this sequences clustered into the ITP receptor clade (Fig. 8). In addition, a membrane guanylyl cyclase (GC-II) is considered as the receptor activated by MIH, resulting in immediate increase of the intracellular messenger cGMP level (Kim et al., 2004; Lee et al., 2007). Therefore, conclusive identification of the CHH/MIH receptor awaits a functional assay that shows CHH/MIH activation of their putative receptor expressed in a heterologous reporting system (Aizen et al., 2016; Ventura et al., 2017).

Leucine-rich repeat-containing GPCRs (LGRs)

As a member of rhodopsin-like GPCR family, three distinct types of LGRs have been classified (A-C) based on the number of leucine-rich repeat (LRR)motifs, low density lipoprotein (LDL) motifs and hinge region sequence(Van Hiel et al., 2012). Four putative LGRs were identified in the phylogenetic analysis (Fig. 8). Pt-GPCR-A11 and Pt-GPCR-A14 clustered with the burscion receptor (type B LGR2) and Pt-GPCR-A16 clustered with GPA2/GPB5 receptor (type A LGR1). LGR2 serves as the receptor for burscion, which regulates molting of the cuticle in C. maenas (Webster et al., 2013; Wilcockson & Webster, 2008) and reproduction and ovarian development in female shrimp, P. monodon (Sathapondecha, Panyim & Udomkit, 2015). In D.melanogaster, GPA2 and GPB5 function via DLGR1 (Hsueh et al., 2005), which has been reported to play a critical role in development (Vandersmissen et al., 2014). The function of LGR1 remains a little obscure in crustacean species. Another LGR identified (Pt-GPCR-A33) clustered with relaxin receptor (LGR4,type C1 LGR). Recent experimental evidence from D. melanogaster suggests LGR4 activated by dilp7, which has also been called relaxin (Imambocus et al., 2020). In insects, relaxin influences lipid synthesis and regulates egg laying decisions (Semaniuk et al., 2018; Yang et al., 2008). In 2020, Vennstra (Veenstra, 2020) has been reported relaxin and LGR4 in eight decapod species. It might suggest that relaxin may be LGR4 ligand. Furthermore, ligand–receptor binding tests are needed to verify this hypothesis. However, the physiological function of relaxin in crustaceans remains unclear.

Orphan rhodopsin-like GPCRs

Orphan receptors are predicted to be involved in neuropeptide signaling pathways, but the ligands and function are still unknown. Since seven Drosophila orphan receptors were identified, orphan receptors orthologs have been identified in crustaceans subsequently (Bao et al., 2020; Caers et al., 2012; Johnson et al., 2021). Two putative GPCRs (Pt-GPCR-A12 and Pt-GPCR-A13) clustered with the Moody receptor ortholog and Pt-GPCR-A27 clustered with Tre receptor ortholog. In P. argus, Moody receptors might be indirectly involved in chemical sensing (Kozma et al., 2020). Little is known about Tre receptor phylogeny or function in crustaceans. In D. melanogaster, Tre receptor might have been involved in chemical sensing and in regulating male courtship behavior (Luu et al., 2016). Much experimental evidence are needed to investigate the similar function of Tre receptor in decapod species as those in insects.

Other GPCRs of interest

Two alternatively splices ETHRs (ETHR-A and ETHR-B) are encoded by ethr gene and expressed in discrete central neurons that are thought to be differently involved in pre-ecdysis and ecdysis (Diao et al., 2016; Kim et al., 2006; Žitňan et al., 2007). Pt-GPCR-A6 clustered with ETHR-B ortholog. In crustacean, numerous reports show ETHR transcripts in the CNS, but information on the role of these receptors in molting is still unclear (Minh Nhut et al., 2020). Crz, a paralog of gonadotropin-releasing hormone, has various functions in different insects, such as affecting ecdysis in Manduca sexta (Žitňan et al., 2007), participating in the pigmentation process in locusts (Tawfik et al., 1999) and increasing heart activity in Periplaneta Americana (Veenstra, 1989). However, Crz has no effect on heart activity, blood glucose level, lipid mobilization or pigment distribution in C. maenas (Alexander et al., 2018). Although CrzR is highly expressed in the YO, but it has little effect on ecdysteroid biosynthesis except a slightly stimulation in early postmolt (Alexander et al., 2018; Tran et al., 2019). In addition, Pt-GPCR-A3 clustered with CrzR ortholog. Crz and CrzR have been identified as an orthologue of GnRH and GnRHR in vertebrates, respectively. GnRH has been involved in regulation of reproductive processes in vertebrates. Still, whether Crz/CrzR signaling pathways in crustaceans plays a role in reproduction requires further investigation. Based on two conserved cysteines and an amidated histidine residue at the C terminus, this peptide has been called CCHamide, which is an invertebrate neuropeptide (Roller et al., 2008). In D. melanogaster, CCHamide-2 binds CHHamide-2 receptor to affect feeding behavior (Nitabach Michael et al., 2013). In Gecarcinus lateralis, two putative CCHamide receptors expressed great differences in all four molt stages and its suggest that these receptors might be play an important role in molting (Tran et al., 2019). In addition, no studies reported CCHamide and their receptors related to reproduction in crustaceans. The role of CCHamide and their receptors in reproduction is still vague.

B-family GPCRs

A total of eight predicted GPCRs clustered with Secretin-like receptors (class B) (Fig. 9). PDFR, DH31R, and DH44R are members of B-family GPCRs in insects (Cardoso, Felix & Power, 2014), and all these B-family GPCRs were identified in P. trituberculatus. Pt-GPCR-B1 and Pt-GPCR-B2 clustered with the Calcitonin receptor ortholog. Pt-GPCR-B3, Pt-GPCR-B7 and Pt-GPCR-B8 clustered with PDF receptor ortholog. Pt-GPCR-B4 clustered with DH31 receptor ortholog. Pt-GPCR-B5 clustered with the DH44 receptor ortholog. Pt-GPCR-B6 clustered with the PTH-like receptor ortholog. Notably, three putative PDFRs (Pt-GPCR-B3, Pt-GPCR-B7, and Pt-GPCR-B8) were identified in P. trituberculatus. Also, three putative PDFRs (PDHRs) have been identified in C. maenas (Oliphant et al., 2018). Subsequently, only two of these putative receptors turned out to be functional by ligand–receptor binding tests (Alexander et al., 2020). In addition, Pt-GPCR-B3 is not a complete ORF and that may lead to an inaccurate cluster. Therefore, a complete ORF of Pt-GPCR-B3 should be cloned for reliable identification.

Tissue distribution of neuropeptides and putative GPCRs transcripts using RT-PCR

After obtaining nucleotide sequences of P. trituberculatus neuropeptide, their expression was explored in different tissues using RT-PCR to confirm that neuropeptide transcripts were synthesized in the CNS, and/or ovary; and check whether these transcripts were expressed in other tissues. A total of 32 putative neuropeptide–encoding transcripts of interest were detected in various tissues (Fig. 10). RT-PCR results showed that all of the 32 neuropeptide transcripts were present in CNS, mainly in cerebral ganglia, eyestalk ganglia, and ventral ganglia, which is consistent with their roles as neuropeptides. This finding showed that the predicted neuropeptides sequences, obtained from P. trituberculatus CNS transcriptome databases are reliable. Moreover, twenty-three neuropeptide transcripts, including ACP, ALP, AST-B, AST-CCC, CCAP, CHH1, CHH2, CNM, Elevenin, GPA2, GPB5, GSE, Kinin, Myosuppressin, NP1, NP2, NP3, sNPF, PDH1, PDH2, RYamide, Tachykinin, and Trissin were expressed in the ovary. This suggests that the putative neuropeptide might be involved in ovarian development and ovarian maturation.

Figure 10 Tissue distribution of P. trituberculatus neuropeptide transcripts.

Thirty-two neuropeptide transcripts were detected in the nine tissues from female P.  trituberculatus. M: DNA Marker; CG, cerebral ganglia; EG, eyestalk ganglia; Gi, gill; Hp, hepatopancreas; Ht, heart; Ms, muscle; Ov, ovary; VG, ventral ganalia; YO, Y-organ; N, negative control (representing no template in PCR). β-actin as the reference gene.

A total of 33 neuropeptide GPCR transcripts (27 were Pt-GPCR-As and 6 belonged to Pt-GPCR-Bs) were chosen for assessment of their tissue distribution of female P. trituberculatus (Fig. 11). Most of these neuropeptide GPCRs were expressed in various tissues as well as neuropeptides. Notably, most of the GPCRs found in the ovary were similar to the expression profiles of neuropeptides. This might suggest that neuropeptide/GPCR signaling pathways are involved in regulation of the ovarian cycle. Five GPCRs, including CCAPR, FMRFamide receptor, Proctolin receptor, AST-BR, and PDFRs, were expressed in ovary, and showed similar expression profiles to those in S. paramamosain (Bao et al., 2018b). Other GPCRs expression profiles in P. trituberculatus did not show corresponding GPCRs in S. paramamosain. This might be because the GPCRs selected for assessment of tissue expression were different from those in S. paramamosain. In addition, some unknown GPCRs were detected in the ovaries of two species. Therefore, further studies should explore expression of these transcripts among various species. Pt-GPCR-A9, was highly expressed in gill, but this sequences clustered with the unknown GPCR orthologs. Pt-GPCR-A4 (CCAPR), Pt-GPCR-A27 (FMRFamide receptor)and Pt-GPCR-B6 (PTH-like receptor) were significantly expressed in ovary, compared with the levels in other tissues. Pt-GPCR-A33 (relaxin receptor) was only expressed in CNS and ovary. These results suggest that CCAPR, FMRFamide receptor, PTH-like receptor and relaxin receptor might be related to ovarian development. Further gene expression analysis should be designed to examine the expression of these GPCRs throughout development and in response to processes such as ovarian development and reproductive maturation.

Figure 11 Tissue distribution of P. trituberculatus neuropeptide GPCR transcripts.

Thirty-three neuropeptide GPCR transcripts were detected in nine tissues from female P. trituberculatus. M: DNA Marker; CG, cerebral ganglia; EG, eyestalk ganglia; Gi, gill; Hp, hepatopancreas; Ht, heart; Ms, muscle; Ov, ovary; VG, ventral ganalia; YO, Y-organ; N, negative control (representing no template in PCR). β-actin is the reference gene.

In summary, several neuropeptide/GPCR signaling pathways have been reported in P. trituberculatus as well as other decapod species and in insects. Recent studies report that several neuropeptide/GPCR signaling pathways are involved in crustacean reproduction. These include AST-C/putative AST-CR (Liu et al., 2019a), sNPF/sNPFR (Bao et al., 2018a), and Vasopressin/putative vasopressin receptor (Lin, Wei & Ye, 2020). These signaling pathways were also identified in P. trituberculatus, however information on the function of these pathways is still unclear. Therefore, further studies are required to gain a better understanding of the function of these neuropeptide/GPCR signaling pathways on the reproduction system of decapod crustacean species.

Conclusion

In the present study, 47 neuropeptide transcripts and 47 GPCR transcripts were identified in CNS transcriptome data of P. trituberculatus. Notably, only allatotropin, EH, and FLRFamide were absent in the dataset. A total of 32 neuropeptide transcripts and 33 GPCR transcripts were chosen for assessment of expression in different tissues. Analysis showed that 23 neuropeptide transcripts and 20 GPCR transcripts, were expressed in the ovary implying that they might be involved in regulation of reproduction. Moreover, most of these neuropeptide/GPCR signaling pathways have been reported in P. trituberculatus as well as other decapod species or insects, implying that they may play similar function in other species. Further, studies should explore the functions of these neuropeptides. In summary, the findings of this study provide a reference for exploring neuropeptide/GPCR distribution in other decapod species. In addition, these findings provide a basis for further studies on neuropeptidergic control of the physiology and behavior of decapod crustaceans.

Supplemental Information

Supplemental Information 1 All GPCR sequences used in this study

Click here for additional data file.

Supplemental Information 2 The specific primers used for RT-PCR

Click here for additional data file.

Supplemental Information 3 P. trituberculatus cerebral ganglia, eyestalk ganglia and thoracic ganglia illumina sequencing and transcriptome assembly statistics

Click here for additional data file.

Supplemental Information 4 Deduced amino acid sequences of neuropeptide and neurohormone precursors from Portunus trituberculatus

Click here for additional data file.

Supplemental Information 5 Raw data of neuropeptide genes for RT-PCR

RT-PCR results in Fig. 10 of the main text, red lines represent cropping lines. All gels have been run under the same experimental conditions.

Click here for additional data file.

Supplemental Information 6 Raw data of GPCR genes for RT-PCR

RT-PCR results in Fig. 11 of the main text, red lines represent cropping lines. All gels have been run under the same experimental conditions.

Click here for additional data file.

Supplemental Information 7 Eclosion hormone transcript and amino acid sequences of peptide precursor proteins deduced from Portunus trituberculatus testis transcriptomic data

Click here for additional data file.

Additional Information and Declarations

Competing Interests

Author Contributions

Data Availability

The authors declare there are no competing interests.

Shisheng Tu performed the experiments, analyzed the data, prepared figures and/or tables, and approved the final draft.

Rui Xu performed the experiments, authored or reviewed drafts of the paper, and approved the final draft.

Mengen Wang performed the experiments, prepared figures and/or tables, and approved the final draft.

Xi Xie analyzed the data, authored or reviewed drafts of the paper, and approved the final draft.

Chenchang Bao conceived and designed the experiments, analyzed the data, prepared figures and/or tables, authored or reviewed drafts of the paper, and approved the final draft.

Dongfa Zhu conceived and designed the experiments, authored or reviewed drafts of the paper, and approved the final draft.

The following information was supplied regarding data availability:

The raw data is available at NCBI SRA: PRJNA707143.

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
