# Peer review of "Identification and characterization of expression profiles of neuropeptides and their GPCRs in the swimming crab, Portunus trituberculatus"

_PeerJ, doi:10.7717/peerj.12179_

## Round 0.1 · original submission · Major Revisions

In addition to the comments and suggestions from reviewers, I think this manuscript would greatly benefit from a more in-depth discussion, especially on the phylogenetic aspects of GPCRs.

Reviewers have requested that you cite specific references. You may add them you believe they are especially relevant. However, I do not expect you to include these citations, and if you do not include them, this will not influence my decision.

Reviewer 1 ·

Basic reporting

Well organized and written manuscript. Figures are good quality and clarity.

Experimental design

Very solid experimental design. Replication of transcriptomes and tissue expression patterns is sufficient for a qualitative analysis (identification of transcripts), which is what is done in this manuscript.

Validity of the findings

The finding are well supported by supplemental data. Phylogenetic analysis identifying homologues is strong. High quality figures support the data.

Additional comments

The manuscript is a basic description of peptides and peptide GPCRs in a single species of decapod crustacean, the swimming crab, Portunus trituberculatus.

Strengths include strong experimental design, clear writing and figures. Includes sequences and patterns of expression in various neural and other tissues.

The text does not stray far from the empirical descriptions. Discussion is short, without much phylogenetic analysis save identification of homologues.

Two relevant references not cited are the following. They are relevant and should help with the discussion. For example, both papers identify considerably more GPCRs (especially type A GPCRs) in a cumulative five species, compared with the number of GPCRs identified in the swimming crab. While the number of species studied so far is relatively small, might the lower number of transcripts in swimming crabs compared to other species be real, and if so, is it significant? :
Tran, N.M., Mykles, D.L., Elizur, A. et al. Characterization of G-protein coupled receptors from the blackback land crab Gecarcinus lateralis Y organ transcriptome over the molt cycle. BMC Genomics 20, 74 (2019). https://doi.org/10.1186/s12864-018-5363-9
Rump, M.T., Kozma, M.T., Pawar, S.D., Derby, C.D. G protein-coupled receptors as candidates for modulation and activation of the chemical senses in decapod crustaceans. PLoS ONE 16(6), e0252066 (2021). https://doi.org/10.1371/journal.pone.0252066

Also, the authors might consider submitting their databases to CrustyBase, an interactive online database for crustacean transcriptomes ...: https://crustybase.org/ and Hyde, C.J., Fitzgibbon, Q.P., Elizur, A. et al. CrustyBase: an interactive online database for crustacean transcriptomes. BMC Genomics 21, 637 (2020). https://doi.org/10.1186/s12864-020-07063-2
Minor comments:
In all the figures, the inset (not the legend) misspells two words: “singal” (should be “signal”) and “cleveage” (should be “cleavage”)
5. Confidential notes to the editor

Reviewer 2 ·

Basic reporting

The manuscript is also well written. Clear and relatively professionl English used throughout.

Experimental design

This paper reports on the identification and characterization of expression profiles of neuropeptides and their GPCRs in the swimming crab, Portunus trituberculatus. In general, the work presented in this paper is relatively large, and has been performed to relatively good standard.

Validity of the findings

To identify the neuropeptides and their GPCRs may help to understand some physiology and behavior, such as gonad maturation and molting of decapod crustaceans.

Additional comments

1. In this paper, the author did not verify the results of the transcriptome assembly. Because there are some errors existed in sequences of genes obtained by the Next Generation Sequencing (NGS), it may be better and necessary to detect the accurate sequences of the screened neuropeptides and their GPCRs using normal PCR and sequencing.
2. L52: Which six species of GPCRs have been reported? Please list.
3. The format of the references are not consistent, such as the abbreviation of the name of the journals.

Reviewer 3 ·

Basic reporting

The manuscript has been written in good, unambiguous English (in the main)- I have some comments to make regarding style, which should be helpful. References are fairly comprehensive, but rather incomplete, and often incorrectly cited, which is a failure of scholarship. Several transcriptomes of portunid crustaceans have now been published, and since Portunus trituberculatus is also a portunid, these MUST be referenced and referred to in the text. In particular the omission of Oliphant et al (2018) BMC Genomics 19:711 is unfortunate, since this gave a reasonably complete neurotranscriptome of a closely related species. The presentation of the Figs. and Tables in the text is fine (a few repetitive typographical errors though (see my comments). Sharing/availability of raw data. I have a real problem here. Obviously the conceptually translated sequences are available, but the raw data is not. The transcriptome reported must be firstly deposited at DDBJ/EMBL/GenBank. The raw read sequence files should be archived in the NCBI SRA archive under a Bioproject number. Finally, it would be most useful if Trinotate annotation reports were included in some supplementary files- others may want to mine these for nucleotide as well as conceptually translated sequences. The results are self-contained and relevant, but there is a problem with the descriptions given, which often include references (and not always the most appropriate ones!). As the results section stands, it has a rather unusual structure- whilst being inevitably descriptive, there is also an element of discussion. I would strongly suggest that the authors completely rewrite this section as a “Results and Discussion” section. This would be a real improvement, and make the manuscript much more readable.

Experimental design

The research question is relevant, and reasonably well defined. The methods are described in sufficient detail, and the pipelines used to extract ands assemble transcriptomes are now pretty well defined in quite a number of papers. I do, however, have a problem with the PCR experiments (results on Figs 10, 11). The authors have performed many PCRs on cDNA extracted from a good variety of tissues. However, there are no RT controls! These are essential. It is extremely difficult to remove gDNA contamination, and certainly just doing a Trizol extraction and DNase treatment will never be sufficient- normally further purification to extract mRNA is needed, and since touchdown PCR was used (35 cycles), I wouldn’t be at all surprised to find that some of the fainter bands might well represent gDNA contamination. Also please show the DNA marker size (ladder) on the figures!

Validity of the findings

The data provided, findings, and the analysis seem reasonable in the main. It is good to see that there is pretty good congruence between the neuropeptide families, diversity within transcripts (where a single transcript encodes several different, but related neuropeptides), putative receptor identities in the published decapod (and particularly crab) transcriptomes. I was surprised to see that there were no eclosion hormone-like transcripts. There are two of these in decapod crustaceans, one is only expressed in the eyestalk, and is undoubtedly a neurohormone, whilst the other is expressed just about everywhere. Similarly, I was surprised to see that no AST CC transcripts were found, or the four PDHs which seem to be found in most crabs. The authors refer to one peptide as “ETH”. This seems unlikely, since it is found in nervous tissues, whereas the authentic ETH of insects is found in non-neural tissues, ie the Inka cells.. There is, as yet, no experimental evidence to suggest that this peptide has any role in molting, except that its expression changes dramatically over the molt cycle. Since it has a PRL-amide motif, and seems to be identical in all decapods (unlike the situation in insects where a wide variety of structures are found), it would be more prudent to call it “Carcikinin” for the time being!

Additional comments

Some suggestions and things that caught my eye.

Line 17. Tone down. I suggest “…their receptors that might be involved in regulation of reproductive processes….”
Line 36 include molt-inhibiting hormone.
Line 37 and throughout. Always include a space before the reference brackets.
Line 42. The Tang et al reference refers to the draft genome of P. trituberculatus, so should go in the preceding line. Also, end this sentence with …. “are unknown”.
Line 50…do not share as high an overall…
Line 52. I think this needs a separate emphasis. Many putative neuropeptide receptors have been tentatively identified on the basis of homology, but only a handful have been functionally deorphaned (using heterologous cell-based reporter assays) in crustaceans. Here reference should be made to these.
Line 59. Please include the Carcinus maenas transcriptome reference (Oliphant et al, 2018).
Line 68 … no studies have reported the …
Line 72-74. I think this is a rather sweeping statement. Tone it down.
Line 79 Molt stage should be given.
Line 82 Use the correct term. Ventral ganglion, not thoracic ganglion (the ventral ganglion is fused sub-oesophageal, thoracic and abdominal ganglia in crabs).
Line 86… was removed using…
Line 100 Peptides
Line 101. The reader should be able to easily understand this: one transcriptome was assembled from all the ganglia combined.
Line 121 (and elsewhere). The sequences obtained were from transcripts, not gene sequences, so avoid using this word- I do appreciate that many do though, but I don’t like it.
Line 134. The ovarian stage (s) (i.e.1-4) should be mentioned here.
Line 159… including cerebral, eyestalk and ventral ganglia.
Results section. The structure of this (as mentioned earlier) is rather unusual- I strongly suggest that the results section is combined with the discussion. This would be much easier to do, and is generally used in transcriptome papers, to keep the style concise, yet informative. Additionally, it is vital that references include those where the hormone was first discovered in crustaceans (for example, bursicon, DH31, MIH, CHH splice variants etc).
Line 270. As mentioned earlier, please avoid calling this peptide ETH, its role in molt control in crustaceans hasn’t yet been established.
Line 329…. other crustaceans. Also line 373. And give references here.
Line 377… completely conserved
Line 410. I see what is meant here, but without MS analysis it can’t be said with certainty whether these conceptually translated peptides are sulfated
Line 426 P. clarkii. Also please be more inclusive- include other transcriptomes with Trissin annotated.
Line 438 Clumsy sentence.
Lines 479-481. This doesn’t make sense!
Line 493 delete insect…. Present in related crustaceans, such as…
Line 531. This might suggest…. Again, tone down.
Line 533…. And showed similar…
Line 553. Relaxin in a peptide that is included in the gonadulin peptide family. So far, the roles of gonadulins in insect reproduction remain a little obscure. Veenstra (2020) certainly did not implicate insulin related peptides in crustacean reproduction! Please be more accurate here, otherwise these sweeping statements will mislead those not familiar with the field.
Line 560 onwards. Please use the correct refs here. The Oliphant et al (2018) paper identified three putative PDHRs in C. maenas (and four PDHs) Subsequently only two of these putative receptors turned out to be functional, by deorphaning (Alexander et al, 2019).
Line 571. The Vasopressin (or more correctly inotocin receptor) wasn’t identified in the Lin et al 2020 paper. Again, please be more accurate.
Line 573-575. This sentence is meaningless.
Please correct all figures… Signal peptide, Cleavage site

---

## Round 0.2 · accepted · Accept

The authors did a great job in revising the manuscript. Congratulations!

Reviewer 2 ·

Basic reporting

No comment

Experimental design

No comment

Validity of the findings

No comment

Additional comments

No comment